# Predicting the Presence and Abundance of Bacterial Taxa in Environmental Communities through Flow Cytometric Fingerprinting

Jasmine Heyse,[a] Florian Schattenberg,[b] Peter Rubbens,[c] Susann Müller,[b] Willem Waegeman,[d] Nico Boon,[a] Ruben Props[a]

[a]Center for Microbial Ecology and Technology (CMET), Department of Biochemical and Microbial Technology, Ghent University, Ghent, Belgium
[b]Department of Environmental Microbiology, Helmholtz Centre for Environmental Research–UFZ, Leipzig, Germany
[c]Flanders Marine Institute (VLIZ), InnovOcean site, Ostend, Belgium
[d]KERMIT, Department of Data Analysis and Mathematical Modelling, Ghent University, Ghent, Belgium

**ABSTRACT** Microbiome management research and applications rely on temporally resolved measurements of community composition. Current technologies to assess community composition make use of either cultivation or sequencing of genomic material, which can become time-consuming and/or laborious in case high-throughput measurements are required. Here, using data from a shrimp hatchery as an economically relevant case study, we combined 16S rRNA gene amplicon sequencing and flow cytometry data to develop a computational workflow that allows the prediction of taxon abundances based on flow cytometry measurements. The first stage of our pipeline consists of a classifier to predict the presence or absence of the taxon of interest, with yielded an average accuracy of 88.13% $\pm$ 4.78% across the top 50 operational taxonomic units (OTUs) of our data set. In the second stage, this classifier was combined with a regression model to predict the relative abundances of the taxon of interest, which yielded an average $R^2$ of 0.35 $\pm$ 0.24 across the top 50 OTUs of our data set. Application of the models to flow cytometry time series data showed that the generated models can predict the temporal dynamics of a large fraction of the investigated taxa. Using cell sorting, we validated that the model correctly associates taxa to regions in the cytometric fingerprint, where they are detected using 16S rRNA gene amplicon sequencing. Finally, we applied the approach of our pipeline to two other data sets of microbial ecosystems. This pipeline represents an addition to the expanding toolbox for flow cytometry-based monitoring of bacterial communities and complements the current plating- and marker gene-based methods.

**IMPORTANCE** Monitoring of microbial community composition is crucial for both microbiome management research and applications. Existing technologies, such as plating and amplicon sequencing, can become laborious and expensive when high-throughput measurements are required. In recent years, flow cytometry-based measurements of community diversity have been shown to correlate well with those derived from 16S rRNA gene amplicon sequencing in several aquatic ecosystems, suggesting that there is a link between the taxonomic community composition and phenotypic properties as derived through flow cytometry. Here, we further integrated 16S rRNA gene amplicon sequencing and flow cytometry survey data in order to construct models that enable the prediction of both the presence and the abundances of individual bacterial taxa in mixed communities using flow cytometric fingerprinting. The developed pipeline holds great potential to be integrated into routine monitoring schemes and early warning systems for biotechnological applications.

**KEYWORDS** flow cytometry, 16S rRNA gene amplicon sequencing, cell sorting, machine learning, monitoring, microbial community dynamics, aquaculture

Address correspondence to Nico Boon, Nico.Boon@UGent.be.

**B**acterial communities are complex and highly dynamic associations that play important roles in many biotechnological applications. One issue that hinders efforts to study and manage these communities is the fact that existing technologies to assess community composition either rely on cultivation or necessitate the extraction and sequencing of genomic material, both of which are time-consuming and laborious. As a result, the availability of fine-scale resolution data on bacterial community dynamics is still limited in many fields. One example hereof is the aquaculture sector (1), where the development of effective management strategies to reduce the occurrence of diseases is hampered by our limited knowledge of the microbial ecology of these systems. Additionally, routine monitoring schemes in aquaculture farms still rely mainly on (selective) plating, which prohibits accurate description of general dysbiotic states and specific disease outbreaks.

Flow cytometry (FCM) is a single-cell technique that is increasingly used as a fast and inexpensive tool for characterizing microbial communities in a wide variety of fields, including drinking water production and distribution (2–4), surveys of natural ecosystems (5–8), aquaculture (9), and fermentation (10, 11). Over the last decade, through the development of advanced data analysis pipelines, the application of FCM has moved beyond its initial purpose of estimating cell densities (12). These computational advances include a range of fingerprinting pipelines (13, 14), algorithms for estimating community stability (15), and algorithms for estimating community diversity metrics (16). Flow cytometry-derived diversity metrics have been shown to be highly correlated with those derived from 16S rRNA gene amplicon sequencing in some ecosystems (16–19), suggesting that there is a link between the taxonomic community composition and phenotypic properties as derived through FCM. This observation is supported by the fact that sorted fractions of a community have taxonomic compositions different from that of the entire community (20–24).

Using machine-learning techniques, Bowman et al. (25) and Rubbens et al. (26) showed that the relative abundances of specific operational taxonomic units (OTUs) are predictive of the abundances of high-nucleic acid (HNA) and low-nucleic acid (LNA) subcommunities in FCM data of natural ecosystems, illustrating the possibility of linking specific regions in the cytometric fingerprint to taxonomic groups using modeling approaches. Several studies have sought to further exploit this relationship in order to build predictive models for taxonomic community composition based on FCM data. Most of these studies take a bottom-up approach in which they train predictive models on data of axenic bacterial cultures. Rubbens et al. introduced the use of *in silico* communities based on axenic culture data (27), while Özel Duygan et al. developed a pipeline that allows classification of mixed communities into classes of predefined cell types by comparing data to signatures of a set of strains and bead standards (28). Cytometric fingerprints of axenic cultures are known to be dynamic over time, for example, in the function of growth stage (29–31). Additionally, we have recently shown that the single-cell properties of an individual taxon, as measured by FCM, depend on the presence of other bacterial taxa in the community (32). Hence, a different cytometric signature can be expected for a taxon in a mixed community compared to the one that is observed in axenic culture. Therefore, making predictions based on models that were trained on axenic culture data can sometimes lead to unreliable predictions in mixed communities (32).

In this study, we aimed to further integrate 16S rRNA gene amplicon sequencing and flow cytometry survey data in order to construct models that enable the prediction of both the presence and the abundances of multiple individual bacterial taxa in mixed communities using flow cytometric fingerprinting (Fig. 1). As a case study, we used samples taken from a whiteleg shrimp (*Litopenaeus vannamei*) hatchery, of which the dynamics have previously been described (33). We first verified the taxonomic stratification in the cytometric fingerprints using cell sorting. We then developed a two-stage pipeline using flow cytometry data as the input that, first, predicts the presence/absence of bacterial taxa and, second, predicts the relative abundances of bacterial taxa. Through the direct linking of flow cytometry and amplicon sequencing survey data, the constructed models do not rely on data from axenic cultures. We verified the ability of the models to assign taxa to the specific regions in the cytometric fingerprint using marker gene data from the cell-sorted community fractions

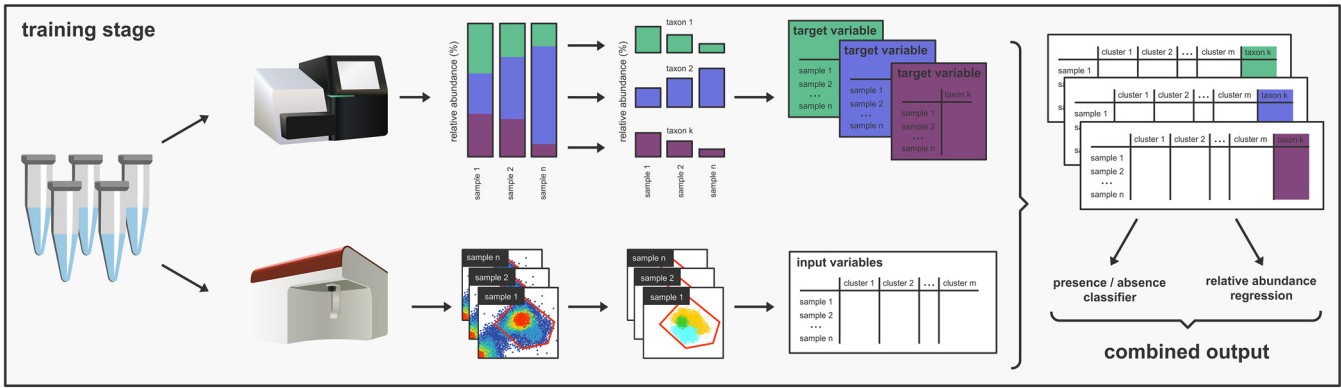

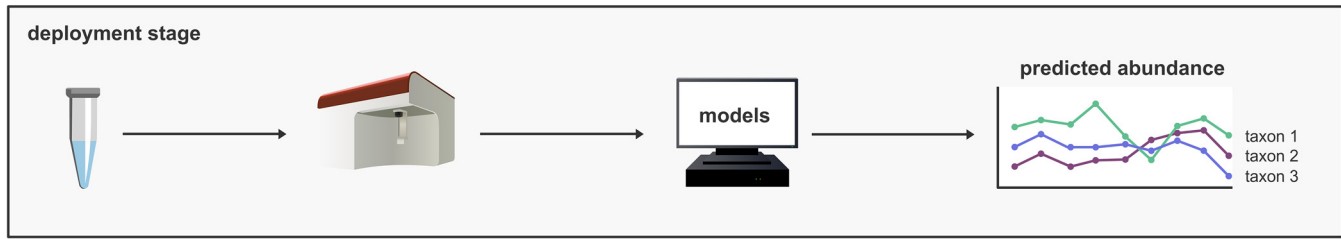

**FIG 1** Overview illustration of the workflow and application of the pipeline presented in this study. During the training stage, samples from the system under study are collected and analyzed using both flow cytometry and 16S rRNA gene amplicon sequencing. For the 16S rRNA gene amplicon data, the reads are processed to calculate relative abundance profiles for each sample. The models are trained for each taxon individually. Therefore, the relative abundances of the taxa of interest are extracted, which results in a single vector for each taxon. For flow cytometry, the single-cell data are separated from the background signals by manually creating a gate on the primary fluorescent channels and subsequently discretized by applying a Gaussian mixture mask, which assigns each cell to a specific cluster. This results in a data frame with the relative abundance for each cluster of the Gaussian mixture in each sample. Two models are constructed for each taxon: an absence/presence classifier and a regression ensemble to predict the relative abundance of the taxon of interest. During the deployment stage, the system under study is sampled using flow cytometry, and the trained models are used to predict the presence/absence and relative abundance of one or multiple taxa of interest.

and using a three-strain mock community. Finally, we tested the approach of our pipeline on two independent data sets to evaluate its applicability to other systems.

## RESULTS

In this study, we used published flow cytometry and 16S rRNA gene amplicon data from an 18-day sampling campaign in an *L. vannamei* hatchery where five replicate cultivations were studied (33). The replicate cultivation tanks were sampled at a resolution of 3 h for flow cytometry and once per day for 16S rRNA gene sequencing. This data set was combined with newly generated 16S rRNA gene amplicon data on sorted fractions of samples originating from this previous study.

**Taxonomic information is conserved in flow cytometric fingerprints.** Prior to model training, the connection between the taxonomic composition of the bacterial communities, as derived through 16S rRNA gene amplicon sequencing, and their phenotypic properties, as derived by flow cytometry, was evaluated using cell sorting. In total, 57 community fractions were sorted from 20 samples using 5 gates (referred to as subcommunities 1 to 5 [SCs 1 to 5]). The sorted regions in the flow cytometry data space (i.e., gates) were chosen to maximize the coverage of the community across the side scatter and SYBR green I fluorescence range (see Fig. S1 at https://doi.org/10.6084/m9.figshare.16337103) and represented subcommunities with relative cell abundances between 3% and 56% of the total cell gate (Fig. 2A).

For all subcommunities, the taxonomic richness was significantly lower than that of the cell gate (one-sided Wilcoxon rank sum test, $P < 0.0001$) (Fig. 2B). The taxonomic compositions of the five gated subcommunities were significantly different from that of the cell gate as well as from each other (permutational multivariate analysis of variance [PERMANOVA] on Bray-Curtis dissimilarities, $P < 0.01$) (see Table S1 and Fig. S2 at https://doi.org/10.6084/m9.figshare.16337103). Each subcommunity was enriched in specific taxa and shared a limited number of taxa with the other subcommunities (Fig. 2C). Many taxa were uniquely detected in a specific subcommunity (e.g., OTU 1 *Phaeodactylibacter* sp. in SC 1); however, some taxa

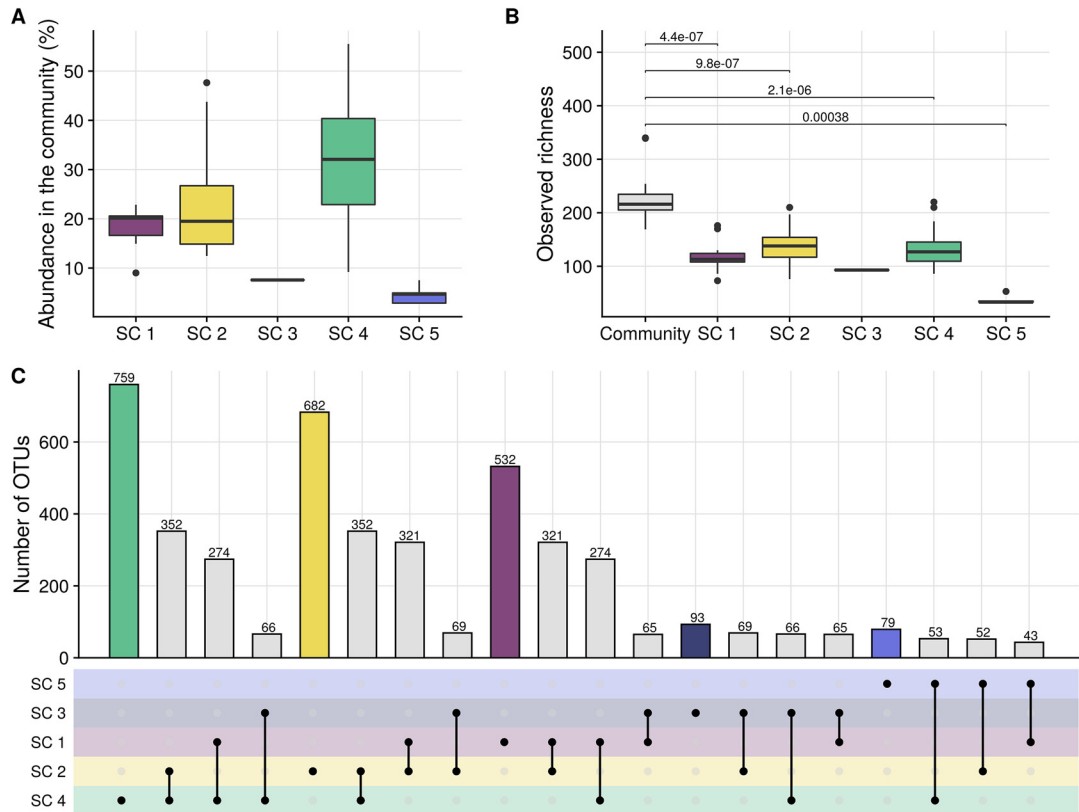

**FIG 2** (A) Relative abundances of the sorted subcommunities (SCs), based on the measurements of the Influx v7 sorter. (B) Observed taxonomic richness in the sorted community and subcommunities. The values above the brackets indicate the *P* values of a one-sided (lower) Wilcoxon rank sum test. Note that for subcommunity 3, no *P* value is supplied since this subcommunity was sorted only once. (C) Upset graph illustrating intersections between the taxonomic compositions of the sorted subcommunities (i.e., the numbers of common OTUs). The upper bars illustrate the cumulative number of OTUs that are found in a subcommunity (in the case of a single dot) or shared between subcommunities (in the case of two connected dots). Note that the numbers of sorted samples were not homogeneously distributed over the five sorting gates (i.e., SCs 3 and 5 were sorted once and three times, respectively, while SCs 1, 2, and 4 were sorted 15, 17, and 18 times [Fig. S2 at https://doi.org/10.6084/m9.figshare.16337103]).

were detected in two (e.g., OTU 3 *Nioella* sp. in SCs 1 and 2) or three (e.g., OTU 7 *Kordia* sp. in SCs 1, 2, and 3) subcommunities (Fig. 2C; Fig. S2 at https://doi.org/10.6084/m9.figshare). The overlap in taxonomic compositions between gates that were more dissimilar from each other in the cytometric space was smaller than that between gates that were more similar (e.g., SCs 1 and 5, which are more dissimilar from each other than from other SCs, share only 15 OTUs, while SCs 1 and 2, which are close to each other, have 147 OTUs in common) (Fig. 2C), confirming that specific taxa typically occur in specific positions of the cytometric space.

The two most narrowly defined subcommunities (i.e., SCs 3 and 5), with the lowest abundance in the community, represented subcommunities with low taxonomic diversity and were nearly mono-dominant, (i.e., *Kordia* sp. in SC 3 and unclassified *Alphaproteobacteria* sp. in SC 5), while the larger and abundant gates (i.e., SCs 1, 2, and 4) were dominated by multiple taxa (Fig. S2 at https://doi.org/10.6084/m9.figshare.16337103). It should be noted that the numbers of sorted samples were not equally distributed over the five sorting gates (i.e., SCs 3 and 5 were sorted once and three times, respectively, while SCs 1, 2, and 4 were sorted 15, 17, and 18 times), which may have caused the cumulative number of observed taxa in SCs 3 and 5 to be lower than in SCs 1, 2, and 4. Nevertheless, also the average number of taxa per sample was lower for SCs 3 and 5 than for SCs 1, 2, and 4 (Fig. 2B).

Throughout the shrimp cultivation, the phylogenetic composition in the subcommunities was preserved well, even though the composition of the total community was dynamic over time and communities differed between the replicate tanks from which samples were sorted.

**Development of a pipeline to extract taxonomic information.** Cell sorting was performed on a different instrument (BD Influx) from that used for the FCM measurements of community samples (BD FACSVerse). To be able to use both the community sample and the sorted sample data as a single data set, a set of representative samples was measured on both instruments, the gates that were used for sorting were manually recreated on the FACSVerse data, and the correspondence between the relative cell abundances in the gates on data of the two instruments was used to evaluate the quality of the manually recreated gates (Fig. S1 at https://doi.org/10.6084/m9.figshare .16337103). The corresponding flow cytometric fingerprints of the sorted subcommunities were obtained from the community measurements using these gates. The combined data set (i.e., including both sorted and community measurements) consisted of 169 samples for which both 16S rRNA gene amplicon and flow cytometry data were available. Models were trained for each OTU individually, using the flow cytometry data as the input and the presence or abundance of the OTU of interest as the model output. Details about the model construction are provided in Materials and Methods. Performances for the top 50 OTUs from the aquaculture data set were evaluated. All reported performance values are performances on the validation sets (i.e., on data that were not used for model training).

In the first part of the pipeline, a presence/absence classifier is trained. Classification performance was evaluated using accuracy and the AUC. The accuracy indicates the percentage of correctly predicted samples in the data set. The AUC indicates the area under the receiver operating characteristic (ROC) curve (i.e., the probability that a randomly chosen sample in which the taxon is "present" is assigned a higher probability for "present" than a randomly chosen sample in which the taxon is "absent"). AUC values allow an easy comparison to a classifier that assigns labels randomly (i.e., AUC = 0.5). We were able to perform presence/absence classification with high accuracies, ranging from 78% to 98% for individual OTUs and AUC values between 0.66 and 0.99 (Fig. 3A and B). The numbers of false-positive samples (i.e., in which the taxon is incorrectly predicted to be present) and false-negative samples (i.e., in which the taxon is incorrectly predicted to be absent) did not differ strongly for individual OTUs (two-sided Wilcoxon rank sum test, $P > 0.05$).

In the second part of the pipeline, the relative abundances of individual taxa were modeled using a regression ensemble. Regression performance was evaluated using $R^2$ i.e., the proportion of the variance in the relative abundance values that can be predicted from the flow cytometry data) and mean average error (MAE), i.e., the average deviation between true and predicted relative abundances). The regression ensembles had $R^2$ values between 0.00 and 0.64 ($0.21 \pm 0.18$ on average) and mean absolute error values between 0.24 and 9.06 ($3.41 \pm 2.19$, on average) (blue dots in Fig. 3). The regression ensembles frequently predicted high relative OTU abundances for samples where an OTU was either absent or present in very low abundance (Fig. S3B at https://doi.org/10.6084/m9.figshare.16337103). Therefore, the predictions of the classifier were superimposed on the regression predictions (Fig. S3A at https://doi.org/10.6084/m9.figshare.16337103); the predicted relative OTU abundances in samples that were classified as absent were set to zero, but predictions for samples in which the OTU was predicted to be present remained unchanged. This reduced the number of false-positive samples by an average of 10-fold (i.e., from $40 \pm 17$ to $4 \pm 3$ out of 100 samples). However, superimposing the classifier to the regression results slightly increased the number of false-negative samples from $3 \pm 3$ out of 100 samples to $8 \pm 5$ on average. Overall, the $R^2$ values were increased to $0.35 \pm 0.24$ on average (ranging between 0.00 and 0.81), and the MAE was reduced to $1.31 \pm 0.97$ on average (green dots in Fig. 3).

To evaluate the ability of our approach to correctly capture dynamics of taxa over time, we predicted the presence and relative abundances of four taxa on the time points for which no amplicon data were available. Additionally, we calculated the predicted absolute OTU abundances by multiplying bacterial densities by the predicted relative OTU abundances. The taxa were selected based on good (OTU 1, $R^2 = 0.81$), intermediate (OTU 2, $R^2 = 0.65$ and OTU 6, $R^2 = 0.19$), and low (OTU 13, $R^2 = 0.03$) overall prediction performances. For OTU 1, the predictions followed the overall patterns that were estimated by interpolation of the time points for

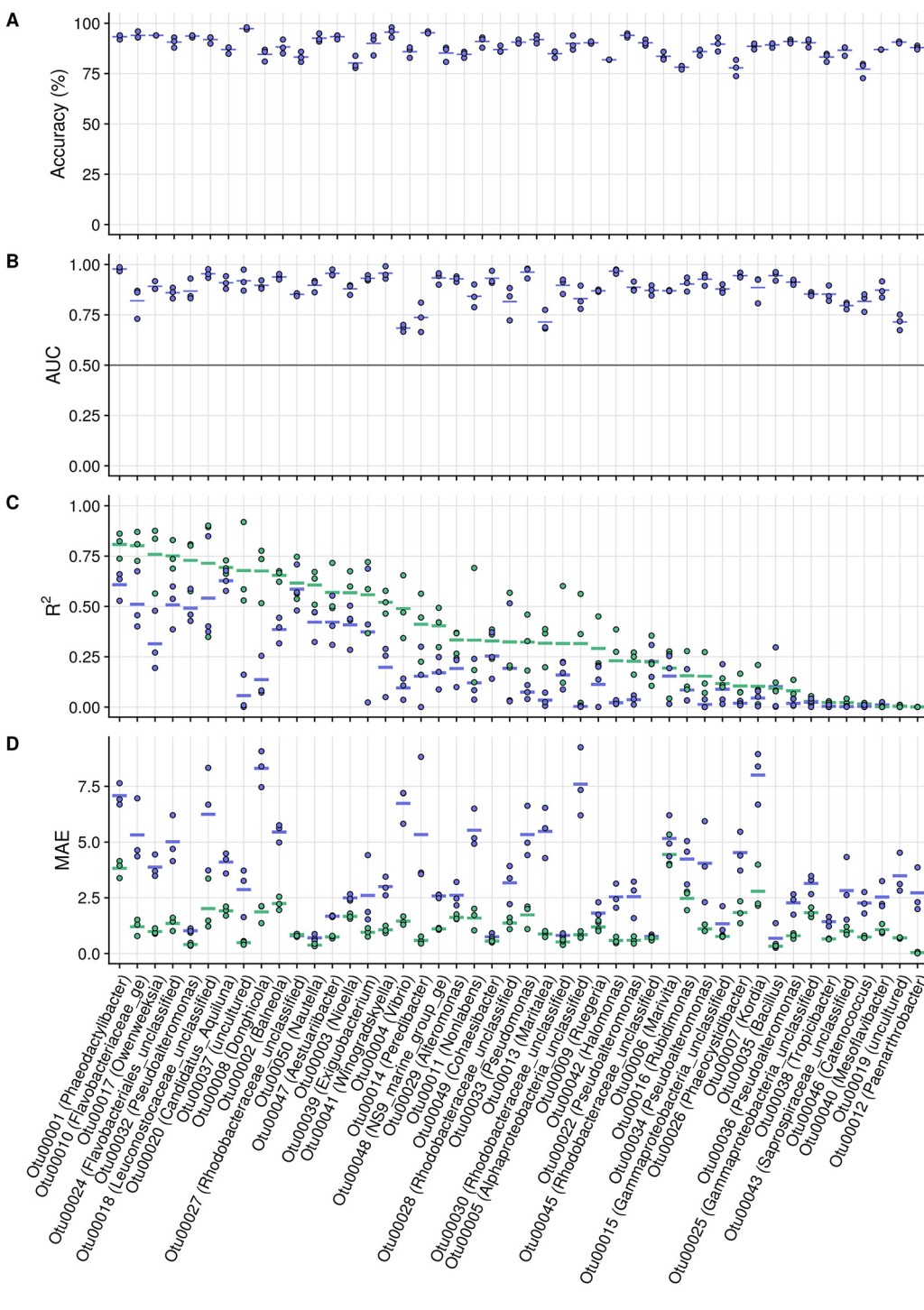

**FIG 3** Classifier accuracy (A), AUC (B), regression $R^2$ (C), and MAE (D) values for the top 50 abundant OTUs from the aquaculture data set. For the regression metrics ($R^2$ and MAE), both the regression model outputs (in blue) and final pipeline outputs (i.e., after imposition of the classifier predictions on the regression results, in green [visualized in Fig. S3 at https://doi.org/10.6084/m9.figshare.16337103]) are illustrated. OTUs are ordered according to their final $R^2$ values. The three dots for each model represent three repeated fold splits, and the vertical line per OTU indicates the average performance of the replicates. The vertical line at 0.5 in panel B indicates the random-guessing threshold of a binary classifier.

which amplicon data were available (Fig. 4). Additionally, the predictions for which the abundances did not match the trends that were estimated by interpolation often coincided with low absolute abundances. Similarly, for OTU 2 and OTU 6, which had intermediate model performances, the abundance patterns followed the expected trends well (Fig. S4 and S5 at https://doi.org/10.6084/m9.figshare.16337103). For OTU 13, which had the lowest performance, the patterns did not correspond to those that were expected based on interpolation of the available data points (Fig. S6 at https://doi.org/10.6084/m9.figshare.16337103).

Since the models were trained on survey data, in which there may be co-occurrence between taxa, predictions of individual OTUs may (partly) rely on detecting co-occurring OTUs and not the OTU of interest itself. In that case, the applicability of the pipeline may be limited to filling gaps in the time series of the data set that was used for model training (i.e., relying on auto-correlation between the samples over time), but the reliability of predictions on independently generated time series of the same environment (e.g., repeated shrimp cultivation in this case) may be limited. To verify the impact of co-occurrence, we compared the performances of models that were trained on only four of the replicate tanks, and predictions were made on the 5th tank (setting 1) with models that were trained using a randomly chosen training and validation set from data of all replicate tanks (setting 2). The former ensured that the co-occurrence patterns of the validation data (i.e., data from the 5th tank) were not incorporated during model training, while the latter incorporated all co-occurrence patterns during model training. There was an average decrease in the $R^2$ of 0.02 across the 50 OTUs in setting 2 relative to that of setting 1. This small decrease suggests that co-occurrence has only a minor influence on model performance. To further investigate how reliably the model can link taxa to a population in the cytometric fingerprint, we assessed, for the top 10 OTUs, the feature importance of the clusters in the cytometric fingerprint (see Materials and Methods for the procedure) with the regions of the sorting gates in which these taxa were observed. Overall, the positions of clusters with high feature importances corresponded well to the positions of the gates in which these taxa were observed, with the exception of OTU 6, for which clusters were detected over the entire range of the bacterial community fingerprint (Fig. S7 at https://doi.org/10.6084/m9.figshare.16337103). For some OTUs there were small deviations, which may be the result of technical aspects. For example, some OTUs were not detected in regions with high feature importances, which may be the result of the limited number of sorted samples and the fact that these were biased toward only 3 tanks during the first half of the sampling campaign (i.e., days 4 to 13). Second, the sorting gates were recreated from the data of one instrument to the other (see Materials and Methods and Fig. S1 at https://doi.org/10.6084/m9.figshare.16337103). This may have caused gates immediately adjacent to the subcommunities to be either marked or not marked, while this was not the case. Overall, these results show that the models can robustly associate taxa with regions in the cytometric fingerprint where they are detected using 16S rRNA gene amplicon sequencing, and hence, they do not rely heavily on co-occurrence patterns.

To test whether taxa that are phylogenetically closely related are more likely to be associated with the same regions in the cytometric fingerprints, the relationship between phylogenetic distance between taxa and feature importance similarity was evaluated. There was a significant (adjusted $R^2 = 0.039$, $P < 2e–16$, Pearson correlation $= -0.20$) relationship between the similarity of cluster importance for different OTUs assigned by the model and the phylogenetic similarities (Fig. S8 at https://doi.org/10.6084/m9.figshare.16337103). This relationship was negative, indicating that OTUs which are phylogenetically closely related are more likely to be associated with the same regions in the cytometric fingerprints.

The sensitivity of the model performance to the amount of data available for training was investigated for two OTUs (i.e., OTUs 1 and 6) by training models on randomly subsampled data sets that contained 20, 40, 60, or 80% of the data set (i.e., 34, 68, 101, or 135 samples). For both of the OTUs and both classification and regression, there was a strong reduction in performance at the lower sample sizes (see the learning curves in Fig. S9 at https://doi.org/10.6084/m9.figshare.16337103). Classification accuracy was reduced by 10% and 5% for OTU 1 and OTU 6, respectively, for every 20% reduction in data set size.

FIG 4 Predictions for OTU 1 (*Phaeodactylibacter* sp.; $R^2$ = 0.81) from the aquaculture data set. The five replicate shrimp cultivation tanks (T1 to T5) were sampled at a resolution of 3 h for flow cytometry and once per day for 16S rRNA gene sequencing. The presence and relative abundances for OTU 1 at the time points for which no amplicon data were available were predicted in order to evaluate the ability of our approach to correctly capture the dynamics of this taxon over time. The dark shades (measured) correspond to the values that were determined based on 16S rRNA sequencing. The lighter shades (predicted) correspond to time points for which only flow cytometry data were available and predictions were made using the models. Expected values can be estimated by interpolation of the measured samples (indicated with the lines between the measured samples). The reported values are averages of the two replicate measurements at each time point. (A) Predictions of the presence/absence classifier. (B) Predicted relative abundances. (C) Predicted absolute abundances, calculated by multiplying the predicted relative abundances by the total cell density as determined through flow cytometry. d, day.

For the regression models, the $R^2$ values were halved when the model was trained on only 20% of the data compared to when it was trained on 80% of the data. For both of the OTUs, the performance did not yet reach a plateau, suggesting that more data are required to improve model performances.

**Application of the approach on external data sets.** To test whether the approach of our pipeline was applicable for monitoring other (managed) microbial systems, the entire workflow was replicated on a three-strain cytometric mock community from the work of Cichocki et al. (34) and a data set of insular reactor communities from the work of Liu et al. (23). Details about the data sets are provided in Table S2 at https://doi.org/10.6084/m9.figshare.16337103.

For the mock community classifier, AUC values ranged between 0.81 and 1.00 and $R^2$ values were 0.89 $\pm$ 0.03, on average (Fig. 5). Since this was a simple mock community, we could validate that the clusters that were assigned a high importance by the model corresponded well to the regions where these taxa were found in the cytometric fingerprint (Fig. S10 at https://doi.org/10.6084/m9.figshare.16337103). For the reactor communities, the AUCs of the top 18 OTUs ranged between 0.57 and 1.00. As for the aquaculture data set, there were big differences in the model performances of individual OTUs. The ranges of performances were similar, as for the aquaculture data set, with an average $R^2$ of 0.33 $\pm$ 0.27.

## DISCUSSION

**Predictive models can link taxa to specific regions in the cytometric fingerprint and predict temporal abundance dynamics.** Substantial variations in model performances were observed for the individual OTUs, for both the aquaculture (Fig. 3) and the validation (Fig. 5) data sets. For all OTUs, the classifier AUC values were largely above the random-guessing threshold of 0.5, indicating that the presence of all taxa could be predicted with moderate-to-high certainty. In contrast, for the prediction of relative abundances, there were large differences in performance between OTUs. For the aquaculture data set, predictions for OTUs with a high-to-intermediate $R^2$ occasionally diverged from what were expected based on interpolation of the time points for which 16S rRNA data were available, but the overall patterns of taxon presence and abundance were predicted well (Fig. 4; Fig. S4 and S5 at https://doi.org/10.6084/m9.figshare.16337103). Based on these results, we conclude that the constructed models are suitable for monitoring dynamics over time but that one should be more cautious when evaluating single snapshot samples. The number of required samples to predict reliable trends depends on the taxa of interest and the dynamics of the system under study. We acknowledge that for a subset of the investigated OTUs, performances were very low and predictions did not correspond to the expected patterns (Fig. S6 at https://doi.org/10.6084/m9.figshare.16337103). Further improvement of prediction performances would greatly increase the applicability of the model. The required model accuracy and tolerated bias will depend on the final context and application (e.g., research, environmental monitoring, pathogen monitoring, etc.). Aspects that can further improve model performances include increased data set sizes for model training (Fig. S9 at https://doi.org/10.6084/m9.figshare.16337103), optimization of acquisition settings, and included fluorescence detectors (35) or the incorporation of different or additional stains in the cytometric measurements (36, 37).

It should be noted that we do not expect the models to improve until the relative abundances of all taxa in a mixed community can be perfectly predicted, since flow cytometric data contain only information regarding a limited set of phenotypic properties. Studies using axenic culture data have observed that some combinations of taxa are difficult to distinguish (28, 38), and studies using sorting and subsequent sequencing typically also observe subcommunities that contain multiple taxa (22). Some taxa may be indistinguishable based on their cytometric fingerprints. Our results indicated that OTUs that are phylogenetically closely related to each other are more likely to be associated with the same regions in the cytometric fingerprints and can therefore be harder to distinguish (Fig. S8 at https://doi.org/10.6084/m9.figshare.16337103). Additionally, some taxa are known to exhibit high phenotypic plasticity (39), which may make it difficult for the model to reliably associate a region in the cytometric fingerprint to such taxa. This implies that we can expect that

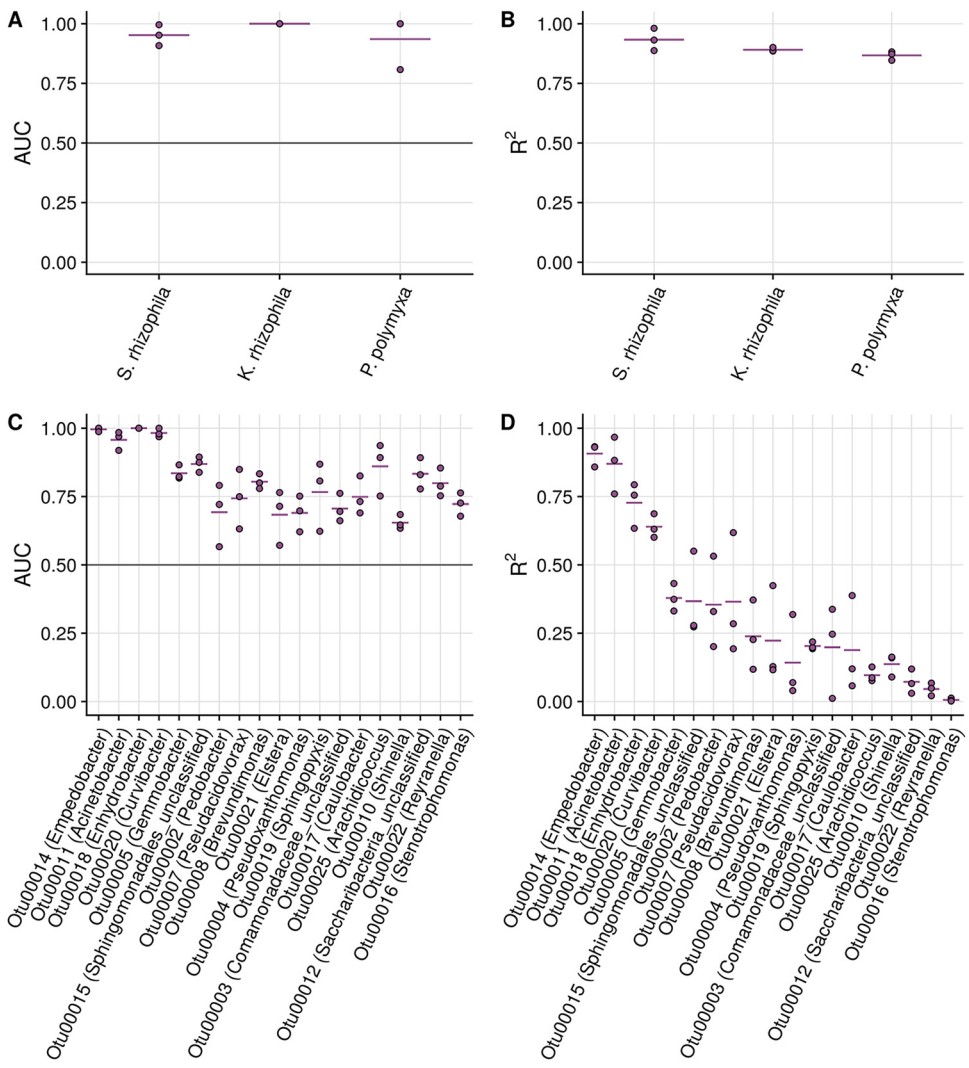

**FIG 5** Model performances with the two validation sets. (A) Classifier AUC values for the three-strain mock community. (B) $R^2$ values for the three-strain mock community. (C) Classifier AUC values for the top 18 OTUs of the reactor communities. (D) $R^2$ values for the top 18 OTUs of the reactor communities. The three dots for each model represent three repeated fold splits, and the vertical line per OTU indicates the average performance of the replicates. The vertical line at 0.5 in panels A and C indicate the random-guessing threshold of a binary classifier.

for some taxa in a given environment, it may be impossible to construct performant models, despite the availability of large data sets and/or sorting data.

In contrast to previously developed methods that predict taxon abundances based on flow cytometry (28, 38), our method does not rely on data from axenic cultures. We have previously shown that the cytometric fingerprint of an individual taxon in the presence of other taxa is different from that in axenic culture and that relative abundance predictions that rely on axenic culture data can therefore be unreliable for mixed communities (32). Hence, for methods that rely on FCM fingerprints of individual taxa for model training, it is recommended that one use experimental setups where it is possible to determine whether the fingerprint in axenic culture is representative of the *in situ* phenotypic fingerprint of the taxon (e.g., through physical separation of cultivated taxa, cell sorting, etc.). By training models directly on survey data of mixed communities, the need to determine the correct *in situ* fingerprint of a taxon prior to model training is removed. We have shown that the models that were constructed for the top 10 OTUs were capable of determining the regions in the cytometric fingerprint that correspond to the locations where these taxa were found through sorting and subsequent amplicon sequencing (Fig. S7 at https://doi.org/10.6084/m9.figshare

.16337103), except for OTU 6. It should be noted that large sorting gates were used in the current study, which allows us to verify only whether the model linked the correct region in the fingerprint to each taxon and not whether the exact cell population(s) was identified correctly by the model. Future validation studies based on sorting with more precise gates are needed to further evaluate the sensitivity of the methodology.

It should be noted that the current study did not verify the performances of the constructed models on an independent data set (e.g., a repeated shrimp cultivation).

**Prospects for bacterial monitoring in biotechnological applications.** The flow cytometric toolbox for monitoring environmental communities already contains algorithms for estimating community-level diversity (16, 40), stability (15), and turnover (41), as well as algorithms that allow one to associate population dynamics with environmental or experimental parameters (13) and pipelines that are designed for community-level classification into different categories (e.g., diseased/healthy, etc.) (42). Standalone community-level metrics, such as diversity or stability, may be difficult to interpret and, therefore, to couple to specific management actions because of the high bacterial heterogeneity and fast dynamics that are typically observed in bacterial systems. Additionally, different shifts in community composition may require a different action of the operator that is monitoring the system. The pipeline of our study allowed us to add an additional layer of taxonomic information to these metrics, which should increase the actionability of operators. Once the models have been constructed, predictions can be made for multiple taxa simultaneously, allowing monitoring of a large fraction of the bacterial community.

We have shown that the pipeline that was developed in this study can be extrapolated to other applications, including analyses of laboratory mock communities and mixed reactor communities (Fig. 5). Performances for the mock-community strains was high, which was expected due to the lower community complexity. Average model performances on the reactor communities were similar to those of the taxa in the aquaculture communities, despite not including sorting data in the training set. This illustrates that cell sorting data are not an absolute necessity to create performant models. However, including sorting data serves as an important validation step for the constructed models (Fig. S7 at https://doi.org/10.6084/m9.figshare.16337103) and may therefore be recommended when studying a new environment. The current study did not include data of natural environments, such as lake and ocean survey data, and hence, the applicability of the pipeline to these environments should be investigated further.

The main advantages of using flow cytometry for community composition monitoring lies in the speed (i.e., minutes) and the high potential for automation (43, 44), which enables monitoring with high temporal resolution. Additionally, the independence of cultivation is a great advantage for monitoring managed ecosystems, since human-induced stressors, such as disinfection, are known to induce viable but nonculturable (VBNC) states (45). Practical applications of the pipeline can include monitoring the efficacy of management strategies, follow-up outbreaks of unwanted bacterial taxa, monitoring the presence of probiotic strains, etc. We believe that the pipeline that was developed in this study holds great potential to be integrated into routine monitoring schemes and early warning systems for biotechnological applications.

## MATERIALS AND METHODS

**Samples.** In this study, we used a combination of previously published flow cytometry and 16S rRNA gene amplicon data from an *L. vannamei* hatchery (33) and newly generated 16S rRNA gene amplicon data on sorted subcommunities of samples originating from that previous study. This data set is referred to as the aquaculture data set. Five gates were created for cell sorting (see Fig. S1 at https://doi.org/10.6084/m9.figshare.16337103). The gates were chosen to cover the range of SYBR green I fluorescence and side scatter that were observed in the data set. The samples that were selected for sorting were chosen from three of the replicate tanks, over different days, in order to include communities with heterogeneous taxonomic compositions.

**Flow cytometry.** Samples for flow cytometry were fixed with 5 $\mu$l glutaraldehyde (20%, vol/vol) per ml (33). Glutaraldehyde-fixed, SYBR green I-stained community samples were measured with a FACSVerse flow cytometer, and sorting was performed with a BD Influx v7 USB sorter. The procedures for flow cytometric measurements, cell sorting, and control samples accompanying these procedures are outlined in detail in the supplemental Materials and Methods at https://doi.org/10.6084/m9.figshare.16337103.

**Illumina sequencing.** Sequencing of the V3-V4 region of the 16S rRNA gene amplicon was performed on an Illumina MiSeq sequencer. The DNA extraction protocols and details about the sequencing are outlined in the supplemental Materials and Methods at https://doi.org/10.6084/m9.figshare.16337103.

**Data sets.** The applicability of the pipeline was additionally tested on two independent data sets: a synthetic community and a mixed community. The synthetic community data set contained samples of a three-strain mock community (*Stenotrophomonas rhizophila* DSM 14405, *Kocuria rhizophila* DSM 348, and *Paenibacillus polymyxa* DSM 36). The reactor community data set originated from the study of Liu et al. (23). More information regarding the data sets, their processing, and availability is provided in Table S2 at https://doi.org/10.6084/m9.figshare.16337103.

**Data analysis. (i) Flow cytometry analysis.** The flow cytometry data were imported in R (v3.6.3) (46) using the flowCore package (v1.52.1) (47). The data were transformed using the arcsine hyperbolic function (16), and the background of the fingerprints was removed by manually creating a gate on the primary fluorescent channels (Fig. S12 at https://doi.org/10.6084/m9.figshare.16337103).

**(ii) 16S rRNA gene amplicon sequencing analysis.** Raw sequencing reads from the previous study and raw sequencing reads generated in this study were processed together. Analysis was performed with the software package mothur (v.1.42.3) (48). Contigs were created by merging paired-end reads based on the Phred quality score heuristic, and they were aligned to the SILVA v123 database. Sequences that did not correspond to the V3-V4 region as well as sequences that contained ambiguous bases or more than 12 homo-polymers were removed. The aligned sequences were filtered, and sequencing errors were removed using the pre.cluster command. UCHIME was used to removed chimeras (49), and the sequences were clustered in OTUs with 97% similarity with the cluster.split command (average-neighbor algorithm). Since the 97% similarity cut-off was used, it is possible that an OTU contained multiple closely related strains. OTUs were subsequently classified using the SILVA v123 database. The OTU table was further analyzed in R (v3.6.3) (46). OTU abundances were rescaled by calculating their proportions and multiplying them by the minimum sample size present in the data set. Absolute taxon abundances were calculated by multiplication of relative abundances with total bacterial densities, as determined through flow cytometry. The correction with cell counts was performed using the determined cell counts for that specific sample.

**Predictive models. (i) FCM preprocessing.** The data were normalized to the [0,1] interval by dividing each parameter by the maximum observed SYBR green I fluorescence channel (i.e., the targeted channel) intensity value over all samples in the data set (i.e., all samples are normalized using the same value). Next, the flow cytometry data were processed by applying a Gaussian mixture mask to the data that allows one to classify each cell into one of the cell clusters that are detected in the data set. For generating the mask, all samples are subsampled to the same number of cells per sample in order to not bias model training toward a specific sample. As with the method of Ludwig et al. (50), the Gaussian mixture model (GMM) was optimized based on the Bayesian information criterion (BIC) using PhenoGMM (19). This discretization results in a one-dimensional (1D) vector for each sample that represents the number of cells present in each mixture. Unless indicated otherwise, the parameters that are included in the model are those that were optimized prior to measurement (i.e., forward scatter [FSC], side scatter [SSC], fluorescence 1 [FL1] [527/32], and FL3 [700/54]). Finally, the mixture counts were converted to relative abundances per sample and transformed using a centered log ratio (clr) transformation implemented in the compositions package (v. 2.0.0) (51):

$$\text{clr}(x_1) = \left[ \frac{x_1}{\left( \prod_{j=1}^{n} x_j \right)^{1/n}} \right]$$

The clr transformation does not change data dimensionality and has previously been used for processing of microbial composition data (52).

**(ii) Illumina preprocessing.** Taxa with low relative abundances are not expected to be detected through flow cytometry. Hammes et al. determined a quantification limit for flow cytometry of $10^2$ cells/ml (53). Since all samples were diluted 10 times, taxa with an absolute abundance below $10^3$ cells/ml were not expected to be observable in the flow cytometry data. Therefore, in each sample, the relative abundance of OTUs with an absolute abundance lower than $10^3$ cells/ml was set to zero.

**(iii) Model training and validation.** For each OTU, a presence/absence classifier and a regressor for relative abundance predictions were trained. To test the robustness of the pipeline, prediction performance was evaluated using independent validation sets with a nested cross-validation scheme (i.e., in the outer loop, 20% of the data are held out for validation of the final model, and in the inner loop, 5-fold cross-validation is used for tuning and training of the models). This outer loop was repeated three times with different fold splits. The pipeline consists of a random-forest (RF) classifier to predict the presence or absence of the taxon of interest and a regression ensemble (i.e., a combination of a gradient boost regression and a support vector regression with polynomial kernel) to predict the relative abundance of the taxon of interest. All models were implemented using the caret (v6.0.86) (54) and caretEnsemble (v2.0.1) (55) packages.

Sequencing survey data are typically zero-inflated (i.e., for each individual OTU, the OTU will be absent or have a very low relative abundance [Fig. S11A at https://doi.org/10.6084/m9.figshare.16337103]). Prior to model training, samples were randomly combined *in silico* to increase the number of samples where the OTU was abundant (Fig. S11B and C at https://doi.org/10.6084/m9.figshare.16337103). This increased model performances (Fig. S11D at https://doi.org/10.6084/m9.figshare.16337103).

For the presence/absence classifier, samples with an OTU abundance lower than 1% were labeled as absent, and samples with an OTU abundance higher than 1% were labeled as present. The reason why

an arbitrary value of 1% was chosen as a cutoff is that small differences in sequencing depth between samples may cause samples with similarly low relative abundances to be labeled differently (i.e., as absent or present). An RF classifier was trained to separate both classes. Before training the classifier, the number of features was reduced using a recursive feature elimination (rfe) strategy (rfe function in caret, 25 iterations). In short, the training data are split into a test set and a train set, the model is tuned on the train set, and the features are ranked according to their importance. For each subset of the $S_i$ most important features, the model is trained on the training set and predictions are made on the test set. This procedure was repeated 25 times, and the average performance profile over the different subset sizes was calculated. The performances quickly reached a plateau. To avoid incorporation of redundant features, the features required to reach an accuracy with a maximal deviation of 0.5% of the maximal accuracy were included (pickSizeTolerance function in caret). Inclusion feature selection improves the ability of the model to use features/clusters that are associated with the modeled taxon and not with correlated clusters that may belong to other taxa (Fig. S12 at https://doi.org/10.6084/m9.figshare.16337103).

For predicting the relative abundances, models with unbound outcomes were used. To avoid the generation of predictions outside the [0,1] range, the logit transformation was applied to map the relative abundances of the individual OTUs to values in the [–Inf, Inf] range before training the regression models, as follows:

$$\text{logit}(x_i) = \ln\left(\frac{x_i}{1 - x_i}\right)$$

Zero values were replaced by one-tenth of the smallest nonzero abundance value. The final regression predictions were inversely transformed so that the final predictions were bound to the [0,1] range. A linear regression ensemble was trained using a gradient-boosting regression and a support vector regression with polynomial kernel. Because the regression models were marked by a high frequency of false-positive predictions, the classifier was used to correct the regression output (i.e., predicted abundances of samples for which the classifier predicted "absent" was set to zero [Fig. S3 at https://doi.org/10.6084/m9.figshare.16337103]).

Relative feature importance values of each model were stored to be compared either between taxa or to the sorting data. For the random-forest classifier and gradient-boosting regression, the mean squared error was calculated on the out-of-bag data for each tree, the values of the variable that was tested were randomly shuffled in the out-of-bag sample, and the mean squared error was calculated again. Differences in the mean squared error values were averaged and normalized. For the support vector regression, the relationship between each predictor and the outcome was evaluated by fitting a loess smoother. The $R^2$ statistic was calculated for this model against the intercept-only null model. This number was returned as a relative measure of variable importance.

**Data availability.** Our entire data analysis pipeline is available as an R Markdown document at https://github.com/jeheyse/FCM-16S_PredictiveModelling. Raw FCM data and metadata for the aquaculture data set are available at FlowRepository under accession number FR-FCM-Z3CY. Raw sequence data of the bulk samples originated from a previous study (33) and are available from the NCBI Sequence Read Archive (SRA) under accession number PRJNA637486. Raw sequence data for the control samples and the sorted and the mock communities generated in this study are available from the NCBI SRA under accession number PRJNA691168. The supplemental information can be found at the following link: https://doi.org/10.6084/m9.figshare.16337103.

## ACKNOWLEDGMENTS

We thank Tim Lacoere for the design of the overview figure of this paper, for his advice during the DNA extractions, and for operating the MinION and Frederiek-Maarten Kerckhof for sharing code to analyze the MinION reads.

J.H. is supported by the Flemish Fund for Scientific Research (FWO-Vlaanderen, project 1S80618N). R.P. is supported by a postdoctoral fellowship of the Flemish Fund for Scientific research (FWO-Vlaanderen, project 1221020N). N.B. is supported by the Bijzonder Onderzoeksfonds (BOF) (BOF15/GOA/006) project. W.W. received funding from the Flemish Government under the Onderzoeksprogramma Artificiële Intelligentie (AI) Vlaanderen Program.

J.H., N.B., and R.P. conceived the study. J.H. and R.P. performed the flow cytometry measurements. F.S. and S.M. performed the sorting analysis. J.H. performed DNA extractions and analyzed the data. R.P., P.R., and W.W. advised the data analysis. R.P. and N.B. supervised the findings of this work. J.H. wrote the paper. All authors contributed to the reviewing and editing of the manuscript. The manuscript was approved by all authors.

We declare that there are no conflicts of interest.

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
