## [Reviewer comments · mSystems]

Predicting the presence and abundance of bacterial taxa in environmental communities through flow cytometric fingerprinting

Jasmine Heyse, Florian Schattenberg, Peter Rubbens, Susann Müller, Willem Waegeman, Nico Boon, and Ruben Props

Corresponding Author(s): Nico Boon, Laboratory of Microbial Ecology and Technology (CMET), Ghent University

Review Timeline:

Submission Date:	May 4, 2021
Editorial Decision:	August 3, 2021
Revision Received:	August 22, 2021
Accepted:	September 1, 2021

Editor: Gail Rosen

Reviewer(s): Disclosure of reviewer identity is with reference to reviewer comments included in decision letter(s). The following individuals involved in review of your submission have agreed to reveal their identity: Stefano Amalfitano (Reviewer #2)

Transaction Report:

DOI: <https://doi.org/10.1128/mSystems.00551-21>

August 3, 2021

Prof. Nico Boon
Laboratory of Microbial Ecology and Technology (CMET), Ghent University
Bioscience Engineering, Ghent University
Coupure Links 653
Gent, O-VI 9000
Belgium

Re: mSystems00551-21 (Predicting the presence and abundance of bacterial taxa in environmental communities through flow cytometric fingerprinting)

Dear Prof. Nico Boon:

Thank you for submitting your manuscript to mSystems. We have completed our review and I am pleased to inform you that, in principle, we expect to accept it for publication in mSystems. However, acceptance will not be final until you have adequately addressed the reviewer comments.

The paper's significance is well-received. There just seems to be some issues with how the normalization is performed and some other minor modifications. Please provide a comprehensive response.

Also, there are 10 supplementary files, and the journal asks if there is a way to condense/compile them so as to have fewer files.

Thank you for your patience with this review - I know that it was delayed, and I am happy that it was well-received and can be accepted soon upon your response.

Preparing Revision Guidelines

For complete guidelines on revision requirements for your article type, please see the journal Article Types requirement at <https://journals.asm.org/journal/mSystems/article-types>. **Submissions of a paper that does not conform to mSystems guidelines will delay acceptance of your manuscript.**

Sincerely,

Gail Rosen

Editor, mSystems

Journals Department
Reviewer comments:

Reviewer #1 (Comments for the Author):

Heyse and coworkers study the suitability of a machine-learning algorithm to predict bacterial community changes in a shrimp hatchery pond from flow cytometry data. This types of procedures may be useful to improve the dynamic coverage of community compositional analysis where sequencing approaches can only provide limited time point analysis.

1) The work is a further step in a series of efforts to more optimally use quantitative flow cytometry data and avoid the frustrating time delays plus costs of extensive community analysis by 16S rRNA gene amplicon sequencing. The latter is widely considered as the gold state-of-the-art, but comes with the drawbacks of cost, lack of absolute quantification of strain abundances, long treatment times and, potentially, overkill. The former is fast (almost real-time), quantitative and easy, and can potentially be coupled to further information on the physiological status of bacterial cells in the community. However, it is also widely acknowledged that cells from pure cultures rarely occupy a single defined multi-parametric 'space' in flow cytometry, and, therefore, it is extremely challenging to predict the most-wanted 'who-is-who' from flow cytometry cell data.

Some recent approaches have suggested that one could capture individual population flow cytometry 'signatures' - or even from different physiological conditions ('sub-population physiological signatures') and train algorithms to differentiate these signatures such that they can be recognized on cells within more complex multi-species mixtures. The results of this are

promising, but not ideal (yet) - as many pitfalls need to be overcome. Also, these approaches rely on being able to culture isolates individually in pure culture, which is not always possible or impractical. Alternatively, as the authors follow here, one could use species-enriched fractions from a complex community (obtained from cell-sorting of flow cytometry gates) - characterize these by 16S rRNA gene amplicon sequencing and use them to train a classifier simultaneously. Ideally, when having enough different sample enrichments to compare with, the classifier would be able to differentiate strains in the mixture.

The results in this manuscript show that this process can work quite reliably, at least for a number of strains, whereas for others it doesn't; possibly because these strains display too many different physiological subpopulations in the mixtures that are not being classified individually by the algorithm. All in all, therefore, a nice piece of work - very extensively documented and validated in a variety of aspects.

2) At the global level, I feel it would be justified to tune down the wording in a manuscript a little bit from the 'marketing' level to a clear scientific level. Whether or not this is THE or a promising pipeline, is not so much the point. Rather the science behind it: how can we optimally recognize cells in complex mixtures, requires our attention and should be taken at a neutral objective value. The reason that I am suggesting is that, first of all, a pipeline as proposed here would require first a detailed analysis of the system by itself through 16S rRNA gene amplicon sequencing, plus cell sorted fractions, plus the training and validation, and then would only be useful for that system. Not really practical, unless such a system would be frequently need to be analyzed. However, to me, this is not the point, but rather: what are the current limitations to recognize strains properly. Secondly, the suggested limited usefulness of the pure culture approach (p. 5, l. 96-107) is not really fair, and inherently in contradiction with the current work itself. Any microbiologist will know that there is phenotypic heterogeneity, and that cell physiology is different on different carbon substrates and growth phases. This is not an argument against using axenic cultures; rather an argument that such growth phases and carbon substrates should be included (and flow cytometry specialist can acknowledge that this gives globally different signals). Potentially also co- or mixed cultures produce different cell heterogeneity, but not to the level as suggested from Heyse 2019 with two species only, one of which produced a different fcm signature in mixed culture. I think that is generalizing a bit too much. If this were so, then the approaches here would be doomed, because then no clear uniform signatures could be expected from any of the strains in the mixtures (which does not seem to be the case!). So my suggestion here: describe this as objectively as you can - it is not a marketing effort. Better use the argument that not all strains that compose the mixture are easily cultured, so sorting fractions makes a lot of sense.

4) The 'validation' effort on two independent data sets is not really a validation in my eyes, because these datasets are independently trained and consist of both 16S rRNA gene amplicon data and cell sorted data. I mean - it is nice to know that also here the procedure works well, but they are independent. Validation would mean that you use the trained algorithm from one on the other. Explain this better in lines 430-435.

I am also wondering whether there would be a more convincing way of validation? Is there a way to apply some 'null' model to show that the classifier actually is doing better than just random? I guess that the predictions in Figure 3B and 3C might be just as well with a tool that simply interpolates from the 16S abundance data?

5) Actually, I am surprised that the enriched in the cell sorting is that 'poor' (Figure 2; still containing 50 or more OTUs) and still useful. What was the motivation to use these gates in particular?

6) How were exactly the absolute OTU abundances produced? I understand that they were

corrected on the basis of FCM counts, but from which time point? Do these overlap? Please add to paragraph l. 457-459.

7) I am not sure I totally understand the procedure of the FCM data preprocessing, but from what is describe here, this seems dangerous to me. Authors say (l. 461-463) are normalized by 'dividing each parameter by the maximum SYBR Green I fluorescence channel (i.e. the targeted channel) intensity value over the data set'. What exactly is this 'data set' and what is the 'maximum..channel intensity value'? Normalizing by the maximum value is potentially extremely distorting, because it is extremely sensitive to outliers. For this reason, in gene expression studies (RNAseq) this is replaced by normalizing to a pseudoreference derived from the median expressions across all samples to be analyzed. A normalized FCM dataset (if this consists of a 'sample' replicate) could thus become 'positioned' differently in multidimensional scale when dividing by a maximum value that is accidentally present but not in others. Depending on what is really meant here, I could strongly suggest to recalculate this using different normalization. Otherwise, please formulate this more precisely.

Also, this sentence suggests that other parameter values (e.g., FSC) are also normalized by the SYBR Green I fluorescence? Can this be clarified?

8) l. 477-479: Note that Illumina 'deep' sequencing has a similar problem of not being able to detect low taxa. If I understand correctly, in their study, the authors use some 30,000 reads per sample, meaning a four log-scale for detection. When analyzing 10e6 cells in flow cytometry (again if I read correctly), the authors have a much larger quantitative scale.

9) l. 483: 'Models are trained for each OTU individually'. Perhaps this can be explained better here? It doesn't seem you have samples with unique single OTUs, but only mixtures - so how are data treated or separated to start with individual OTUs? L. 453: Is 97% OTU similarity here really referring to the same strain or could it be possible that one is actually looking at different strains (for example: clades or closely related species) appearing in different reactors at different moments? This multi-strain presence is suggested by quite some metagenomic studies, on wastewater as well as on gut microbiota.

Reviewer #2 (Comments for the Author):

This study offers an attractive workflow to link the microbiome profiling (16S rRNA gene amplicon sequencing) and flow cytometric fingerprinting to predict the occurrence of major bacterial taxa in a mixed aquatic community from an aquaculture setting.

The approach was validated by cell sorting (i.e., comparing the phylogenetic profiles of sorted populations with the that of the total community) and applied to two external dataset.

The methodological approach is built on a solid literature and described in full detail. The statistic part is rather convoluted but fluently readable.

I have some concerns mainly regarding the reproducibility of the proposed pipeline to the environmental conditions found in natural contexts (i.e., other than the managed ones tested herein).

1- In the title and other parts of the main text (e.g., L358-359), it is emphasized that the pipeline is suitable for environmental communities. This was not directly tested and appears only speculative. The approach is proved effective for microbial monitoring of managed systems, as admittedly stated (e.g., L283-284). In my opinion, this should be clear from the title and on, without ambiguities or hiding the true and demonstrated application of the proposed approach. I understand the need to attract a wider audience, but, please, do not hide the requirements to face monitoring issues in aquaculture and engineered/managed systems.

2- Linked to the previous point. Both amplicon seq and flow cyt have well-known detection limits and will have suboptimal performances when approaching those limits. Therefore, it seems unlikely that the presented approach could work properly at the low microbial abundance values found in many natural waters (10^3 - 10^4 cells/ml) or at the high abundance and diversity levels found in wastewaters and hypereutrophic systems. Again, I suggest to link the study focus to the aquaculture sector.

3- L165-168. What is this specific for?

4- Please, clarify the reason why the samples were diluted 10 times (see L477-482). Is this issue affecting the model performances (see e.g., L279-281; L316-318). In general, it is important here to set clearly the applicability limits of the proposed application.

5- I have noted several different data transformation options introduced along the procedure. A large body of the literature is based on other options to treat similar data. To what extent the outcomes of this study are/can be affected by such questionable steps?

6- L297-303. This part is redundant with the aims reported at the end of the introduction. It is unusual to have it at the beginning of the discussion. Moreover, the point 3 is not an aim of the study but one of the supportive activities.

7- L322. =...will depend

Response to reviewers comments

Dear editor,

I am pleased to re-submit the enclosed manuscript entitled “Predicting the presence and abundance of bacterial taxa in environmental communities through flow cytometric fingerprinting” by Jasmine Heyse, Florian Schattenberg, Peter Rubbens, Susann Müller, Willem Waegeman, Nico Boon, Ruben Props for publication in the *mSystems* journal.

We are thankful for the detailed comments provided by the reviewers. Comments have been addressed and we indicated where the changes in the revised manuscript were made. Supplementary information is posted on figshare (<https://doi.org/10.6084/m9.figshare.16337103>).

Reviewer 1

We thank the reviewer for the detailed summary of our results and conclusions, constructive comments and the general positive statement about our work. We have rephrased several parts of our manuscript in order to clarify some aspects or to better position our results.

- 1. At the global level, I feel it would be justified to tune down the wording in a manuscript a little bit from the 'marketing' level to a clear scientific level. Whether or not this is THE or a promising pipeline, is not so much the point. Rather the science behind it: how can we optimally recognize cells in complex mixtures, requires our attention and should be taken at a neutral objective value. The reason that I am suggesting is that, first of all, a pipeline as proposed here would require first a detailed analysis of the system by itself through 16S rRNA gene amplicon sequencing, plus cell sorted fractions, plus the training and validation, and then would only be useful for that system. Not really practical, unless such a system would be frequently need to be analysed. However, to me, this is not the point, but rather: what are the current limitations to recognize strains properly.**

We agree with this point and have made several adjustments throughout the text (mainly in the discussion) to address this comment. Additionally we reduced the length of the second part in the discussion section to position the pipeline relative to the set of existing flow cytometry tools for monitoring diversity/stability/etc., and to focus less on the practical applications of the pipeline.

To apply the here described pipeline a thorough dataset consisting of joint 16S rRNA gene amplicon sequencing and flow cytometric data is required. The use of cell sorting is however not an absolute requirement to be able to construct performant models, but rather serves as an important validation step. For example, in our second validation set (the “reactor dataset”) only community samples were included. To make this more explicitly clear to the readers we have added a comment regarding this in the discussion (lines 382-388).

- 2. Secondly, the suggested limited usefulness of the pure culture approach (p. 5, l. 96-107) is not really fair, and inherently in contradiction with the current work itself. Any microbiologist will know that there is phenotypic heterogeneity, and that cell physiology is different on different carbon substrates and growth phases. This is not an argument against using axenic cultures; rather an argument that such growth phases and carbon substrates should be included (and flow**

cytometry specialist can acknowledge that this gives globally different signals). Potentially also co- or mixed cultures produce different cell heterogeneity, but not to the level as suggested from Heyse 2019 with two species only, one of which produced a different fcm signature in mixed culture. I think that is generalizing a bit too much. If this were so, then the approaches here would be doomed, because then no clear uniform signatures could be expected from any of the strains in the mixtures (which does not seem to be the case!). So my suggestion here: describe this as objectively as you can - it is not a marketing effort. Better use the argument that not all strains that compose the mixture are easily cultured, so sorting fractions makes a lot of sense.

Thank you for this comment, we are in agreement with this reasoning. We made modifications to the text to better clarify that our approach does not constitute a direct argument against methods based on axenic culture model training.

However, the interpretation that is made regarding implications of our previous study (Heyse *et al.*, 2019) is not complete. It is important to note that we found that when taxa are cultivated together, both the position and variance of each population's cytometric cell signatures differed from when they were grown in axenic cultures. This was observed for both taxa, but to a different extent (Figure 1A & B below; created with figures from our previous study). These results do not imply that no clear uniform signatures will be expected from strains in a mixed community, as proposed in the question, but rather that inter-taxon interactions can cause taxa to adopt different cytometric signatures. In our previous study we therefore tested how this affected predictions for a pipeline that was relying patterns from axenic culture data and found that models that are trained on axenic data give different abundance predictions as compared to models that were trained on data from physically separated cocultures (Figure 1C below; figure from our previous study). As such, we cannot assume that models that are constructed on data of axenic cultures will always allow to make accurate predictions in mixed communities. This observation does not contradict the current study, but rather motivates to try to develop predictive models that are trained directly on survey data for environments where inter-taxon interactions play a significant role. When using survey data as a starting point the exact location of a taxon in the cytometric fingerprint does not need to be known prior to model construction.

We want to stress that we do not argue against the use of models that rely on axenic culture data, but we suggested that they should be used in cases where it is possible to verify that the fingerprint of the axenic culture is representative enough for the *in situ* fingerprint of taxa (e.g. through sorting) or when the fingerprint of the taxon was determined in a setup that is representative for the system for which predictions will be made (e.g. through the use of membrane-separated coculture systems, etc.; see Discussion lines 340-355). Additionally, we have rephrased some sentences in the introduction to make the implications of different fingerprints under coculture conditions more clear to a reader that is not familiar with our previous study, and to make it more clear that the influence of these different fingerprints on the predictions was observed and is not an assumption (lines 105-107).

Figure 1 – Assembled figure with illustrations from Heyse et al. (2019). (A) Combined figure of the experimental set-up (Figure 1) and the cytometric fingerprints at 72h for *Enterobacter* sp. in axenic culture and as part of a coculture (Supplementary Figure 5), illustrating the shift in fingerprint during cocultivation as compared to axenic cultivation. (B) Combined figure of the experimental set-up (Figure 1) and the cytometric fingerprints at 72h for *Pseudomonas* sp. in axenic culture and as part of a coculture (Supplementary Figure 5), illustrating the shift in fingerprint during cocultivation as compared to axenic cultivation. (C) Predictions that were obtained when a model was trained based on pure culture data as compared to when a model was trained on coculture data (Supplementary Figure 10).

3. The 'validation' effort on two independent data sets is not really a validation in my eyes, because these datasets are independently trained and consist of both 16S rRNA gene amplicon data and cell sorted data. I mean - it is nice to know that also here the procedure works well, but they are independent. Validation would mean that you use the trained algorithm from one on the other. Explain this better in lines 430-435.

Correct, this indeed may cause confusion among readers, as the term validation is used interchangeably in model training and model application. The “validation” in that section refers to the validation of the methodological approach on other datasets/environments and not a validation of the constructed models itself. We have adjusted the formulation in the text (lines 283-296) to more clearly indicate that only an application of the approach on other systems was included and not a real model validation and added a comment in the discussion regarding the need for further validating the constructed models itself on other datasets (lines 361-362).

Ideally, future model validation should include two steps, i.e. in a first part the constructed models could be applied on an independent dataset coming from a similar ecosystem (e.g. a shrimp cultivation), and in the second phase the constructed models could be applied on a dataset coming from a different environment

where similar community members are found (e.g. since the models were constructed on a dataset for marine aquaculture, a dataset of marine monitoring campaign, etc.). Prediction accuracies in this second set-up can be expected to be lower as compared to the first set-up. Performing this validation in two steps should allow us to get an idea on how easily models can be extrapolated to other environments and will provide more information regarding how flexible the position of taxa in cytometric fingerprints are over different communities/environments.

4. I am also wondering whether there would be a more convincing way of validation? Is there a way to apply some 'null' model to show that the classifier actually is doing better than just random? I guess that the predictions in Figure 3B and 3C might be just as well with a tool that simply interpolates from the 16S abundance data?

Comparing classifier performance to random can be done based on the obtained area under the ROC values (AUC). The AUC value indicates the probability that a randomly-chosen sample where the taxon is “present” is assigned a higher probability for “present” than a randomly-chosen sample where the taxon is “absent”. An AUC of 0.5 indicates that the classifier does not assign a higher probability for “present” than for “absent” to samples that are in reality labelled as “present”. In that case, the classifier cannot distinguish between “absent” and “present” and it will classify the samples randomly. The obtained AUC values in our dataset were ranging between 0.66 and 0.99 for the complete aquaculture dataset (Figure 3B in the manuscript), 0.81-1.00 for the mock community (line 289), and 0.57-1.00 for the reactor community (line 294). Hence, all constructed models were performing significantly better than random. In the revised version of the manuscript we have replaced the average AUC value with the range of observed AUC values for the mock and reactor communities to make it more clear than AUC is never ≤ 0.5 .

In our manuscript we decided to report both accuracy and AUC-values for the classifiers. Accuracy values illustrate how frequently we expect the model to predict the correct label if the model would be used on unseen data, which is easy to interpret and is of interest when the model is being used in practice. This metric does not take into account the class balance, and therefore does not allow to estimate whether the model is performing better than random. The AUC values allow to compare performance to random, and, therefore, give a better idea on how good the constructed model is able to distinguish the samples that are labelled as “present” and “absent”. However, the AUC value is more difficult to interpret with respect to the expected frequency of misclassifications. We have added a sentence to explain better what both accuracy and AUC re indicating for evaluating classifier performance (line 186-192). During the review we noticed there was a small mistake in Figure 3 in the revised manuscript, i.e. the caption it states that an accuracy of 50% is corresponding to a random classifier. This would only be correct if the classes were perfectly balanced, which is not the case. The line at 0.5 for AUC is correct.

For the question regarding interpolation from the relative or absolute abundance data: interpolation can be an easy solution dependent on the application. If the intended application of the model is to make predictions in new datasets from the same environment (e.g. investigate a repeated shrimp cultivation) the 16S data would only be acquired in the first survey, the models would be constructed on this data, and from then on the models can be used to make predictions for samples from that same environment. During the use of the model there would not be any 16S amplicon data to interpolate from. However, if the application of the models would be restricted to filling up gaps in a dataset for which on some timepoints both FCM and 16S amplicon data is available and on other timepoints only FCM data is available, interpolation may indeed be sufficient for some research questions. However, some dynamics that are typically observed in bacterial communities will not be captured through linear interpolation. An example

hereof, is the wide range of non-linear growth dynamics that is observed for bacterial taxa, including logistic growth, patterns of diauxic growth, etc.

5. Actually, I am surprised that the enriched in the cell sorting is that 'poor' (Figure 2; still containing 50 or more OTUs) and still useful. What was the motivation to use these gates in particular?

We decided to choose the gates in order to cover most of the cytometric fingerprint of the communities without splitting cell clusters. Hence, the gates varied in size, and some of them were relatively large which resulted in most subcommunities to contain a range of taxa, rather than selecting very narrow gates that are highly enriched in only one or a few taxa. The motivation for this choice was that we wanted to be able to estimate the location for most of the abundant taxa in the cytometric fingerprint, since we wanted to use this information for validating whether our models were able to find the locations where different taxa occur (Fig S7 at <https://doi.org/10.6084/m9.figshare.16337103>). The instrument we used was capable of performing 4-way sorting (i.e. maximally 4 sorted populations could be obtained per sample) and we intended to sort the same populations multiple times over the duration of the sampling campaign and over the different tanks in order to be able to confirm that the locations of the different taxonomic group remained preserved in our dataset. If we would have decided to focus our sorting effort on narrow gates with more highly enriched populations, this would have allowed us to either only investigate a limited number of narrowly defined cytometric populations (that had been sorted multiple times), or to investigate a higher number of narrowly defined cytometric populations (that had been sorted a limited number of times). Both of these options seemed less suitable for our current study as compared to choosing larger gates. However, the use of such smaller gates would be interesting for further work since it would allow to link some taxa to very specific 'populations' in the fingerprint (whereas we can now only verify whether the model is able to link taxa to 'regions' in the fingerprint). Additionally, including such data during model training can be expected to increase model performances for these taxa. We have added a comment about the added value of sorting with narrow gates for future studies (lines 358-360).

6. How were exactly the absolute OTU abundances produced? I understand that they were corrected on the basis of FCM counts, but from which time point? Do these overlap? Please add to paragraph I. 457-459.

The FCM-based counts of the rearing water communities were obtained during our previous study (Heyse *et al.*, 2021), and were available for each tank and at each timepoint for which also sequencing data was available. Hence the correction with cell counts was always performed using the determined cell counts for that specific sample. We have clarified this better in the revised manuscript (lines 452-454).

7. I am not sure I totally understand the procedure of the FCM data preprocessing, but from what is describe here, this seems dangerous to me. Authors say (I. 461-463) are normalized by 'dividing each parameter by the maximum SYBR Green I fluorescence channel (i.e. the targeted channel) intensity value over the data set'. What exactly is this 'data set' and what is the 'maximum..channel intensity value'? Normalizing by the maximum value is potentially extremely distorting, because it is extremely sensitive to outliers. For this reason, in gene expression studies (RNAseq) this is replaced by normalizing to a pseudoreference derived from the median expressions across all samples to be analyzed. A normalized FCM dataset (if this consists of a 'sample' replicate) could thus become 'positioned' differently in multidimensional scale when dividing by a maximum value that is accidentally present but not in others. Depending on what is really meant here, I could strongly suggest to recalculate this using different normalization. Otherwise, please formulate this more precisely. Also, this sentence suggests that other

parameter values (e.g., FSC) are also normalized by the SYBR Green I fluorescence? Can this be clarified?

In this section the "dataset" that is referred to contains all FCM samples included in the study and the "maximum channel intensity value" is the maximal value in the channel that picked up the signal from SYBR Green I. Hence, during this normalization, one single value is used for the normalization of all samples. Therefore, the relative positioning of cells over the different samples will not be influenced. The presence of outliers do not pose a problem during this procedure since we performed a gating on the bacterial population at the start of the sample processing. After the gating no extreme outliers are present in the data anymore. This is illustrated in Figure 2A, B & C (below). This normalisation procedure is common practice in FCM fingerprinting pipelines. As suggested, we have rephrased the section where the normalization is explained in the manuscript in order to avoid confusion regarding which value is used (lines 456-459).

It is correct that all parameters are divided by the maximal SYBR Green I fluorescence value. The reasoning behind this is that the SYBR Green I channel is the targeted channel and hence the highest intensity values are expected to be observed in that channel. If the normalization on the other channels is performed using the highest SYBR Green I value, normalized values in other channels are still in the [0,1] range (illustrated in Figure 2D below). An advantage of using the same value is that also the relative spread along the different parameters will be preserved.

Figure 2 – Illustration of the normalization that is performed. (A) The original data of a random sample of the FCM prior to gating. (B) The same sample after gating (when outliers are no longer present). (C) The data after normalization by dividing each parameter by the maximum SYBR Green I fluorescence intensity value that was observed over all samples from the dataset. (D) The scatter

values on data that was normalized using the maximum SYBR Green I fluorescence intensity value that was observed over all samples from the dataset.

- 8. I. 477-479: Note that Illumina 'deep' sequencing has a similar problem of not being able to detect low taxa. If I understand correctly, in their study, the authors use some 30,000 reads per sample, meaning a four log-scale for detection. When analysing 10e6 cells in flow cytometry (again if I read correctly), the authors have a much larger quantitative scale.**

This is an excellent remark. The two technologies indeed have a different dynamic reach. In the current study we have not yet investigated the limits of detection and the potential implications of having different limits of detection for the both technologies. To investigate this some additional research would be required. For example, from previous studies, we used a detection limit for flow cytometry of 10^2 cells/mL has been presented (Hammes et al., 2008). Since all samples were diluted 10 times, taxa with an absolute abundance below 10^3 cells/mL were not expected to be observable in the flow cytometry data. Hence we set the relative abundance of those taxa to zero (lines 475-480). However, we expect that in practice this limit might be more complex, as the detection of the cells by the FCM does not imply it can be distinguished and quantified by the model. Similarly, in our current study we have used an arbitrary 1% cut-off in the Illumina data for the absence/presence classifier (lines 498-502). Further testing and optimisation of this cut-off in relation to the sensitivities of the two technologies would be needed to get a better idea of the dynamic range of the models.

- 9. I. 483: 'Models are trained for each OTU individually'. Perhaps this can be explained better here? It doesn't seem you have samples with unique single OTUs, but only mixtures - so how are data treated or separated to start with individual OTUs? L. 453: Is 97% OTU similarity here really referring to the same strain or could it be possible that one is actually looking at different strains (for example: clades or closely related species) appearing in different reactors at different moments? This multi-strain presence is suggested by quite some metagenomic studies, on wastewater as well as on gut microbiota.**

We mean that for each OTU, a classifier and a regressor is constructed separately (Figure 3 below, Figure 1 in the manuscript). Data from the OTU of interest is extracted from the OTU table, combined with the FCM data (i.e. matched according to which sample they originate from) and this combined dataset is used for model construction. We stated this explicitly here to avoid confusion with compositional or multi-output modelling approaches which would attempt to predict the abundance of all taxa in the OTU table using a single model. We have adjusted this sentence in order to make this statement more clear (lines 481-482).

It is possible that the OTUs that were defined at the 97% similarity level in our study are representing multiple strains, which is a general limitation of 16S rRNA gene amplicon data that we cannot escape. Whether we would be able to distinguish them and predict their abundances individually will need to be investigated further. We have found that phylogenetically more closely related taxa are more likely to be associated with the same regions in the cytometric fingerprints (Fig S8 at <https://doi.org/10.6084/m9.figshare.16337103>). This also points to the fact that distinguishing closely related strains will be difficult and potentially not possible. We added a sentence to indicate that strains may be grouped together in our study (lines 447-448).

Figure 3 – Snippet from Figure 1 in the manuscript, illustrating which datasets were used for model training. For each OTU, the relative abundances were extracted from the OTU table and combined with the FCM samples. This combined dataset was used for model construction.

Reviewer 2

We thank the reviewer for the thorough examination of our manuscript. We have added additional clarification regarding the different data processing steps and regarding the environments that were included in our study.

- 1. In the title and other parts of the main text (e.g., L358-359), it is emphasized that the pipeline is suitable for environmental communities. This was not directly tested and appears only speculative. The approach is proved effective for microbial monitoring of managed systems, as admittedly stated (e.g., L283-284). In my opinion, this should be clear from the title and on, without ambiguities or hiding the true and demonstrated application of the proposed approach. I understand the need to attract a wider audience, but, please, do not hide the requirements to face monitoring issues in aquaculture and engineered/managed systems.**

This is an interesting point. We did not aim to attract wider audiences, but we meant to refer to the applicability of the method for mixed communities that are found in the environment (i.e. not limited to mixed/synthetic communities and laboratory studies). We believe there is a semantic difference in what constitutes an “environmental” community and a “natural” community. By most definitions, the microbial communities in an aquaculture system would classify as environmental communities, given their origin (i.e. open aquatic systems), and composition, but not as a “natural” community. However, it is correct that we did not include such a natural environment such as a lake or marine survey dataset in our study. To address this comment we have made an adjustment in the text to state this explicitly (lines 388-390). However, we have no reason to suspect that training models on data from natural environments would be problematic (see response to question below).

- 2. Linked to the previous point. Both amplicon seq and flow cyt have well-known detection limits and will have suboptimal performances when approaching those limits. Therefore, it seems unlikely that the presented approach could work properly at the low microbial abundance values found in many natural waters (10^3 - 10^4 cells/ml) or at the high abundance and diversity levels found in wastewaters and hypereutrophic systems. Again, I suggest to link the study focus to the aquaculture sector.**

Thank you for this comment. We share your concern on the detection limits connected to our approach, and this has been the main reason why we focused on the most abundant taxa (i.e. top 50 OTUs) in the system, and did not overinterpret the results from the least abundant taxa. While we agree that different systems will differ greatly in their microbial cell density, the aquaculture systems and reactor system used as cases here, do cover $1e5 - 1e7$ cells per mL (a two \log_{10} variation) over the survey data, which is representative of most environmental systems. Additionally, the model uses the relative cluster abundances in the FCM data as features (lines 468-469). Therefore, the absolute abundance of the community is of less concern.

Nevertheless, the question regarding the potential application of the methodology to other environments is interesting and could not yet be addressed in the current study since our study included only datasets from managed environments (i.e. the aquaculture dataset, insular reactor communities and synthetic community). Previous research has shown that the link between cytometric and taxonomic diversity is also found in communities in natural water bodies (e.g. bacterioplankton communities in coastal areas (García et al., 2015), bacterioplankton communities present in two fresh water lake communities (Props et al.,

2018)). Additionally, the studies of Bowman et al. (2017) and Rubbens, Schmidt, et al. (2019) were able to link OTUs to HNA/LNA populations in a marine and lake ecosystem, respectively. These results illustrate that training predictive models to find the link between taxa and their region in the cytometric fingerprint in natural ecosystems will be possible.

We have made some changes to the discussion to emphasize the applicability of the methodology for the studied datasets while stating clearly that our datasets did not contain data of a natural community (lines 388-390).

3. L165-168. What is this specific for?

This is for technical reasons. The BD FACSVerser was available in the lab where all analysis from our previous study were performed (CMET, Ghent University, Belgium). However, this instrument is not equipped with sorting functionality. Therefore, the sorting samples that were obtained for the study were performed at the Department of Environmental Microbiology, UFZ, Germany.

4. Please, clarify the reason why the samples were diluted 10 times (see L477-482). Is this issue affecting the model performances (see e.g., L279-281; L316-318). In general, it is important here to set clearly the applicability limits of the proposed application.

This is for technical reasons. The BD FACSVerser is able to accurately acquire data for maximally 35,000 events per second (reference document: https://www.bdbiosciences.com/content/dam/bdb/marketing-documents/BD_FACSVerser_TechSpecs.pdf). These “events” include the microbial community, background signals from the sample matrix, and some instrumental noise. When the samples of the current study were measured undiluted, the event rate was higher than this threshold, resulting in unreliable data acquisition. In order to avoid this, all samples were diluted 10 times in a sterile buffer. Diluting samples prior to measurement is common practice in microbial flow cytometry analysis.

5. I have noted several different data transformation options introduced along the procedure. A large body of the literature is based on other options to treat similar data. To what extent the outcomes of this study are/can be affected by such questionable steps?

There are indeed several steps in the data processing. These are either common practice to analyse these type of data, or, when they are less common, the effect of these on the of the pipeline has been verified. We have listed each of them below. In order to address this comment added some references in the materials and methods section to illustrate more clearly that each of the data processing steps is either established in other studies or verified during our study (section “Data analysis”).

- Asinh transformation on FCM data (line 435): The channel intensity values for microbial cells are spread across several orders of magnitude. To be able to distinguish signals from cells from the instrument noise, data require to be transformed (see recent reviews regarding data-analysis for FCM: Montante and Brinkman, 2019; Rubbens and Props, 2021). The asinh transformation is frequently used, however, alternative transformations are being used in scientific literature as well.
- Normalisation of FCM data (line 456): During this normalization the intensity values for each parameter are divided by the maximum observed SYBR Green I fluorescence channel (i.e. the targeted channel) intensity value over all samples in the data set (i.e. all samples are normalised using a single value). This only results in a rescaling of the data in the [0,1] interval and hence this

does not influence the final model construction and associated performances. This normalisation is common practice in fingerprinting pipelines such as *Phenoflow* (Props *et al.*, 2016).

- Centred log ratio transform of compositional (FCM) data (lines 470-474): The model uses the relative abundances of the cell clusters that are detected by the GMM as input features (line 468). Many standard statistical methods lose their applicability when they are used on compositional data, since this data is subjected to a unit sum constraint, which makes, for example, correlations spurious to interpret. These issues can be avoided by applying transformations to the compositional data which removes this unit sum constraint, and after which standard statistical models can be used again. Several options for this transformation exist. We were constrained to use a method that does not affect the dimensionality of our data as one of the goals of our study was to be able to verify whether the model was able to detect the clusters in the cytometric fingerprint which corresponded to each taxon based on our sorting results (Fig S8 at <https://doi.org/10.6084/m9.figshare.16337103>). The clr-transformation does not change data dimensionality and has previously been used for processing of microbial composition data (Gloor *et al.*, 2017) and hence this was included.
 - Correction of Illumina data with cell counts (lines 452-545): This transformation consists of multiplying relative abundances of taxa in a sample with the total cell density in that sample. The transformation to absolute taxon abundances was only required for data filtering and was reversed prior to model training. Hence, this did not influence model performances. This correction has been proposed in several studies (Props *et al.*, 2017; Vandeputte *et al.*, 2017).
 - Logit transformation for bound model outcomes (lines 516-520): This is necessary to ensure that the model output will be bound to the [0,1] interval. If no such transformation is included in the data the model could output nonsensical results (e.g., relative abundances more than 100% or less than 0%).
 - *In silico* data mixtures to reduce high zero inflation (lines 493-497): zero inflation is a well-known problem when analysing community composition data. In order to increase the relative proportion of samples where the taxon of interest was present, *in silico* samples consisting of a weighted mix of two lab samples were made. Including this step increased model performances in our study (Fig S11 at <https://doi.org/10.6084/m9.figshare.16337103>). Also intuitively it seems reasonable that making *in silico* combined samples should not bias our models: if we assume that a specific cell cluster in the cytometric fingerprint is corresponding to a specific taxon, and we make a weighted sum of the cluster abundances in two samples, we expect the abundance of the taxon to be the weighted sum of the relative abundances in each of these two samples.
- 6. L297-303. This part is redundant with the aims reported at the end of the introduction. It is unusual to have it at the beginning of the discussion. Moreover, the point 3 is not an aim of the study but one of the supportive activities.**

We have removed the section.

7. L322. =...will depend

This has been adjusted in the revised manuscript.

References

- Bowman, J.S., Amaral-zettler, L.A., Rich, J.J., Luria, C.M., and Ducklow, H.W. (2017) Bacterial community segmentation facilitates the prediction of ecosystem function along the coast of the western Antarctic Peninsula. *ISME J.* **11**: 1460–1471.
- García, F.C., Alonso-sáez, L., Morán, X.A.G., and López-urrutia, Á. (2015) Seasonality in molecular and cytometric diversity of marine bacterioplankton : the re-shuffling of bacterial taxa by vertical mixing. *Environ. Microbiol.* **17**: 4133–4142.
- Gloor, G.B., Macklaim, J.M., Pawlowsky-glahn, V., and Egozcue, J.J. (2017) Microbiome Datasets Are Compositional: And This Is Not Optional. *Front. Microbiol.* **8**: 1–6.
- Hammes, F., Berney, M., Wang, Y., Vital, M., and Egli, T. (2008) Flow-cytometric total bacterial cell counts as a descriptive microbiological parameter for drinking water treatment processes. *Water Res.* **42**: 269–277.
- Heyse, J., Buysschaert, B., Props, R., Rubbens, P., Skirtach, A.G., Waegeman, W., and Boon, N. (2019) Coculturing Bacteria Leads to Reduced Phenotypic Heterogeneities. *Appl. Environ. Microbiol.* **85**: 1–13.
- Heyse, J., Props, R., Kongnuan, P., Schryver, P. De, Rombaut, G., Defoirdt, T., and Boon, N. (2021) Rearing water microbiomes in white leg shrimp (*Litopenaeus vannamei*) larviculture assemble stochastically and are influenced by the microbiomes of live feed products. *Environ. Microbiol.* **23**: 281–298.
- Montante, S. and Brinkman, R.R. (2019) Flow cytometry data analysis: Recent tools and algorithms. *Int. J. Lab. Hematol.* **41**: 56–62.
- Props, R., Kerckhof, F.-M., Rubbens, P., De Vrieze, J., Hernandez-Sanabria, E., Waegeman, W., et al. (2017) Absolute quantification of microbial taxon abundances. *ISME J.* **11**: 584–587.
- Props, R., Monsieurs, P., Mysara, M., Clement, L., and Boon, N. (2016) Measuring the biodiversity of microbial communities by flow cytometry. *Methods Ecol. Evol.* **7**: 1376–1385.
- Props, R., Schmidt, M.L., Heyse, J., Vanderploeg, H.A., Boon, N., and Deneff, V.J. (2018) Flow cytometric monitoring of bacterioplankton phenotypic diversity predicts high population-specific feeding rates by invasive dreissenid mussels. *Environ. Microbiol.* **20**: 521–534.
- Rubbens, P. and Props, R. (2021) Computational Analysis of Microbial Flow Cytometry Data. *mSystems* **6**: 1–12.
- Rubbens, P., Schmidt, M.L., Props, R., Biddanda, B.A., Boon, N., Waegeman, W., and Deneff, V.J. (2019) Randomized Lasso Links Microbial Taxa with Aquatic Functional Groups Inferred from Flow Cytometry. *mSystems* **4**: 1–17.
- Vandeputte, D., Kathagen, G., Kevin, D., Vieira-silva, S., Valles-colomer, M., Sabino, J., et al. (2017) Quantitative microbiome profiling links gut community variation to microbial load. *Nature* **551**: 507–511.

September 1, 2021

Prof. Nico Boon
Laboratory of Microbial Ecology and Technology (CMET), Ghent University
Bioscience Engineering, Ghent University
Coupure Links 653
Gent, O-VI 9000
Belgium

Re: mSystems00551-21R1 (Predicting the presence and abundance of bacterial taxa in environmental communities through flow cytometric fingerprinting)

Dear Prof. Nico Boon:

Congratulations!!

Your manuscript has been accepted, and I am forwarding it to the ASM Journals Department for publication. For your reference, ASM Journals' address is given below. Before it can be scheduled for publication, your manuscript will be checked by the mSystems senior production editor, Ellie Ghatineh, to make sure that all elements meet the technical requirements for publication. She will contact you if anything needs to be revised before copyediting and production can begin. Otherwise, you will be notified when your proofs are ready to be viewed.

As an open-access publication, mSystems receives no financial support from paid subscriptions and depends on authors' prompt payment of publication fees as soon as their articles are accepted. =

Publication Fees:

- Minimum resolution of 1280 x 720
- .mov or .mp4. video format
- Provide video in the highest quality possible, but do not exceed 1080p
- Provide a still/profile picture that is 640 (w) x 720 (h) max

· Provide the script that was used

We recognize that the video files can become quite large, and so to avoid quality loss ASM suggests sending the video file via <https://www.wetransfer.com/>. When you have a final version of the video and the still ready to share, please send it to Ellie Ghatineh at eghatineh@asmusa.org.

Sincerely,

Gail Rosen
Editor, mSystems

Journals Department
Phone: 1-202-942-9338